# Kaempferol-3-O-Glucuronide Ameliorates Non-Alcoholic Steatohepatitis in High-Cholesterol-Diet-Induced Larval Zebrafish and HepG2 Cell Models via Regulating Oxidation Stress

**DOI:** 10.3390/life11050445

**Published:** 2021-05-14

**Authors:** Yang Deng, Ji Ma, Xin Weng, Yuqin Wang, Maoru Li, Tingting Yang, Zhiyang Dou, Zhiqi Yin, Jing Shang

**Affiliations:** 1School of Traditional Chinese Pharmacy, China Pharmaceutical University, Nanjing 211198, China; 1821020454@stu.cpu.edu.cn (Y.D.); 3119020098@stu.cpu.edu.cn (J.M.); 1721020382@stu.cpu.edu.cn (X.W.); 3220020355@stu.cpu.edu.cn (Y.W.); 3119020097@stu.cpu.edu.cn (M.L.); 3320021431@stu.cpu.edu.cn (T.Y.); 2020182112@stu.cpu.edu.cn (Z.D.); 2Jiangsu Key Laboratory of TCM Evaluation and Translational Research, China Pharmaceutical University. Nanjing 211198, China; 3State Key Laboratory of Natural Medicines, China Pharmaceutical University, Nanjing 211198, China

**Keywords:** non-alcoholic fatty liver disease, kaempferol-3-O-glucuronide, oxidative stress, inflammation, high-cholesterol diet.

## Abstract

NAFLD (non-alcoholic fatty liver disease) is one of the most prominent liver diseases in the world. As a metabolic-related disease, the development of NAFLD is closely associated with various degrees of lipid accumulation, oxidation, inflammation, and fibrosis. Ilex chinensis Sims is a form of traditional Chinese medicine which is used to treat bronchitis, burns, pneumonia, ulceration, and chilblains. Kaempferol-3-O-glucuronide (K3O) is a natural chemical present in Ilex chinensis Sims. This study was designed to investigate the antioxidative, fat metabolism-regulating, and anti-inflammatory potential of K3O. A high-cholesterol diet (HCD) was used to establish steatosis in larval zebrafish, whereby 1mM free fatty acid (FFA) was used to induce lipid accumulation in HepG2 cells, while H_2_O_2_ was used to induce oxidative stress in HepG2. The results of this experiment showed that K3O reduced lipid accumulation and the level of reactive oxygen species (ROS) both in vivo (K3O, 40 μM) and in vitro (K3O, 20 μM). Additionally, K3O (40 μM) reduced neutrophil aggregation in vivo. K3O (20 μM) also decreased the level of malondialdehyde (MDA) and significantly increased the level of glutathione peroxidase (GSH-px) in both the HCD-induced larval zebrafish model and H_2_O_2_-exposed HepG2 cells. In the mechanism study, *keap1*, *nrf2*, *tnf-α*, and *il-6* mRNA were all significantly reversed by K3O (20 μM) in zebrafish. Changes in Keap1 and Nrf2 mRNA expression were also detected in H_2_O_2_-exposed HepG2 cells after they were treated with K3O (20 μM). In conclusion, K3O exhibited a reduction in oxidative stress and lipid peroxidation, and this may be related to the Nrf2/Keap1 pathway in the NAFLD larval zebrafish model.

## 1. Introduction

For centuries, natural products have been used as medicinal treatments for the treatment and prevention of disease. Multiple active compounds have been found in naturally occurring products which can be used to treat a myriad of diseases. One such example is a traditional Chinese medication called Ilicis Purpureae Folium, which originates from the dried leaf of Ilex chinensis Sims. Typically, it is used to treat bronchitis, burns, pneumonia, ulceration, and chilblains [1]. Various active compounds have been found in Ilicis Purpureae Folium, including flavonoids, phenolic acids, triterpenoids, and triterpenoid saponins. In addition, several compounds in Ilicis Purpureae Folium have been reported as having antioxidant and anti-inflammatory effects [2,3,4], which means that Ilicis Purpureae Folium is a potential source of active compounds.

Kaempferol-3-O-glucuronide (K3O) attracted our attention due to its existing in both Ilicis Purpureae Folium and Cyclocarya paliurus, which have a liver protecting effect [5,6]. Moreover, recent papers have reported that K3O can be used as an active in vitro antioxidant [7] and an anti-inflammatory compound [8]. Taken together, these discoveries indicated that K3O could be a potential therapeutic drug for related diseases.

Non-alcoholic fatty liver disease (NAFLD), also known as metabolic associated fatty liver disease (MAFLD), is one of the most common liver diseases in the world [9]. However, as the secondary reason for liver transplants, NAFLD has no specific therapeutic drug approved by the FDA [10]. This disease is defined as the abnormal accumulation of lipids in the liver (>5% by weight) in the absence of alcoholic and viral factors. It can be classified into two categories: simple lipid accumulation, in which only steatosis can be observed; and non-alcoholic steatohepatitis (NASH), in which lobular inflammation and liver cell injury can be observed [11]. The pathogenesis of NAFLD is complex and remains unclear to this day. Evidence indicates that the combined effect of some genetic phenotypes such as lipid metabolism, oxidation, and inflammation results in NAFLD [12]. As crucial factors in the occurrence and progression of NAFLD, anti-inflammation and antioxidation may be potential targets for NAFLD. Hence, anti-inflammatory and antioxidant compounds, such as widely researched lipid accumulation reduction compounds, are also reasonable treatment options for NAFLD [13].

The aim of the present work is to investigate whether K3O can effectively prevent NAFLD in larval zebrafish with a HCD diet and HepG2 with FFA- or H_2_O_2_-induced cell injury. Meanwhile, this study aimed to clarify the potential mechanisms by determining several key mRNA expression levels of multiple signaling pathways.

## 2. Materials and Methods

### 2.1. Reagents and Solution

K3O was provided by Zhiqi Yin’s lab at China Pharmaceutical University, the purity of which was greater than 98%. H_2_O_2_ was purchased from Nanjing Chemical Reagents Co., Ltd., and Dihydroethidium (95%), Nile red (95%), Bezafibrate (98%), and Oil Red (98%) were obtained from Aladdin (Shanghai, China). Cholesterol and 2,7-Dichloro-dihydrofluorescein diacetate (DCFH-DA) were purchased from Sig-ma-Aldrich (St. Louis, MO, USA).

### 2.2. Network Pharmacology

Herb-pharmacological activities, herb-compound, and compound-target relations were presented by the online Traditional Chinese Medicine Systems Pharmacology (TCMSP) database, and the analysis platform (http://ibts.hkbu.edu.hk/LSP/tcmsp.php, accessed on 12 May 2021) [14,15,16,17]. The relationship between disease and Ilicis Purpureae Folium is shown in Appendix A. Appendix A presents the related compounds in Ilicis Purpureae Folium, and the compound-target network list of K3O.

### 2.3. Cell Culture and Treatment

Human hepatoma cells HepG2 were cultured in high glucose Dulbecco modified Eagle medium (DMEM) medium supplemented with 10% fetal bovine serum without antibiotics. The culture conditions were 37 °C, and 5% CO_2_ humidified atmosphere. Before the experiments, the surface of the cells was washed with phosphate buffered solution (PBS). The cells were treated in a serum-free medium throughout the en-tire experiment.

Two different models were employed in the experiments: the lipid accumulation model and the oxidative stress model. In the lipid accumulation model, we used 1mM free fatty acid (FFA) (oleic acid, OA: palmitic acid, PA, 2:1)/10% BSA in DMEM in the experiments. When FFA was added to the cells, different levels of K3O were also given. The final concentrations of K3O in the media were 10 μM (K3O-L), 15 μM (K3O-M), and 20 μM (K3O-H). FFA and K3O were co-cultured with the cells for 24 h. Then, after washing the cells with PBS three times, we used the Oil Red working solution mentioned above to cover their surface and stain them for 30 min.

In the oxidative stress model, cells were pretreated with 10 μM (K3O-L), 15 μM (K3O-M), and 20 μM (K3O-H) K3O as the final concentrations for 24 h, and were then washed with PBS. After removing the cell treatments, we added 200 μM H_2_O_2_ into the DMEM medium for 1 h. Then, after washing it with PBS three times, the cells were observed and imaged by a stereoscope. The superoxide level was indicated by the dihydroethidium (DHE, 10 μM) stain level at 490 and 610 nm. The exposure intensity and time for all the photos among the same experiments were consistent.

### 2.4. Zebrafish Body Size Measurements

Using ImageJ software, the width and length of the belly of each zebrafish were semi-quantitatively processed and counted with reference to zebrafish photographs. Ten zebrafish weights were taken as a group and weighed with an analytical balance.

### 2.5. Maintenance of Zebrafish and Treatment

Wild-type AB-line zebrafish and neutrophil-specific transgenic zebrafish Tg (*mpx*: EGFP) were reared in filtered circulating water with a light cycle of 14:10 (light:dark) hours at 28.5 °C. The embryos were produced from wild-type or Tg (*mpx*: EGFP) lines of maturated zebrafish by natural reproduction. The zebrafish embryos were grown freely in egg water with a light cycle of 14:10 (light:dark) hours at 28.5 °C until 5 days post fertilization (dpf). Then, the larval zebrafish were randomly divided into six groups (*n* = 100 for each group).

In the experiments, 5% cholesterol-containing (HCD) basic larval zebrafish food (Gardners, AP, USA) was prepared as previously reported [13]. Dimethyl sulfoxide (DMSO) was used to dissolve the K3O and Bezafibrate (BZT), followed by dilution with water to achieve the final drug concentration of K3O (10 μM, 20 μM, and 40 μM) and BZT (10 μM). The final DMSO concentration was 0.001%.

HCD was given from 6dpf, and compound treatments were given from 8 dpf for the model group and treatment group. Meanwhile, the control group was fed with a standard diet. The experiments were stopped at 21 dpf. All animal experiments were authorized by the Science and Technology Department of Jiangsu Province and followed the Jiangsu Provincial standard ethical guidelines in using experimental animals under the ethical committees mentioned above.

### 2.6. Oil Red Staining and Histopathology

Zebrafish were anesthetized with 0.05% tricaine before all stains. A total of 0.5 mg/mL Oil Red was dissolved with isopropyl alcohol, then diluted with water (3:2, *v:v*) to prepare Oil Red working solution. Furthermore, the Oil Red working solution was filtered until the supernatant was clear and transparent. Meanwhile, 4% paraformaldehyde-fixed zebrafish and cells were stored at 4 °C for 24 h before the experiment. Samples were washed with PBS three times and then soaked in isopropyl alcohol for 5 s. After the sample preparation, the Oil Red working solution was used to stain them. The zebrafish samples were stained for 1 h, while the cells only need to be stained for 30 min. To remove the background of the samples, we employed 80% isopropyl alcohol to do a single wash after staining. The prepared samples were then observed directly under a light microscope (Olympus, Tokyo, Japan). The staining intensity of Oil Red in the obtained images was quantified using the ImageJ software. The hematoxylin and eosin (HE) stains were performed by Microworld Biotech Co. Ltd. (Nanjing, China).

### 2.7. Fluorescent Staining and Quantification

Three different fluorescent stains were used in our experiments: Nile red (Aladdin, China), dichloro-dihydro-fluorescein diacetate (DCFH-DA; Aladdin, China), and dihydroethidium (DHE; Aladdin, China). Nile red is a photostable, lipophilic dying with bright red fluorescence, which is often used in neutral lipid staining. Excitation and scattering wavelength of Nile red were 543 and 598 nm, respectively. DCFH-DA is a cell-permeable probe used to detect reactive oxygen species (ROS). ROS can transform nonfluorescent samples to fluorescent ones by generating dichlorofluorescein (DCF) in living cells, whose excitation and scattering wavelength are 480 and 525 nm, respectively.

In the experiments, zebrafish were maintained in 0.5 μg/ml Nile red or 10 μM/L DCFH-DA for 30 min at 28.5 °C in darkness. After being stained, zebrafish were washed with egg water three times, then fixed with 4% CMC-Na. Nile red or DCFH-DA-stained photos were observed and snapped by a fluorescence stereoscope (Olympus SZX16). The exposure intensity and time corresponding to the same stain were consistent for comparison. The quantifications of Nile red and ROS were obtained by smashing the stained zebrafish (*n* = 10 for each group) and then detecting its fluorescence intensity using a multifunctional microplate reader.

### 2.8. Biochemical Measurement

Each experimental group contained five samples consisting of six zebrafish. Ice water was used for euthanasia, and the samples used for biochemical tests were crushed via ultrasonic soundwaves. Triglycerides (TG), total cholesterol (TC), malondialdehyde (MDA), and GSH-px levels were measured with commercial assay kits (Jiancheng, Nanjing, China) according to the manufacturers’ instructions. The cell biochemical measurements were all carried out following the manufacturers’ instructions. The quantitation results of the above kits were read with a multifunctional microplate reader (BioTek, Winooski, VT, USA).

### 2.9. RT-qPCR

A real-time RT-qPCR was used to determine the expression of the genes involved in lipid metabolism, oxidation, and inflammation. A total of 20 larval zebrafish were taken from each group and euthanized to extract the total Ribonucleic acid (RNA) by applying Trizol reagent (Invitrogen, Waltham， MA, USA). A reverse transcription kit (PrimeScript RT Master Mix, Takara, Japan) was used in the cDNA collecting process. Furthermore, the corresponding mRNA expressions were quantified using qPCR reagent (SYBR Green, Takara, Japan). All steps in the experiments were conducted according to the manufacturers’ protocol in the kits. Oligo in qPCR was purchased from Genscript (Nanjing, China), whose information is shown in Table 1. The expression levels of each targeted mRNA sequence were calculated by applying the 2−ΔΔCt method after being normalized to GAPDH.

### 2.10. Statistical Analysis

Graph Pad PRISM (Graph Pad Software, San Diego, CA, USA) was applied to analyze the data. All data were presented in the form of mean ± SD. One-way ANOVA was used in the significant calculation, while *p* < 0.05 in different groups was regarded as statistically significant.

## 3. Results

### 3.1. Effect of K3O on Body Size on HCD-induced Larval Zebrafish

The body size of the larval zebrafish was affected by HCD, BZT, and K3O. In Figure 1, we compared zebrafish body length, abdominal width, and weight data. The experimental results showed that the HCD group was significantly higher than the normal group in the three types of body size data. Both BZT and K3O were shown to significantly reverse the changes of zebrafish shape induced by HCD, and BZT had the most obvious callback trend. These results suggest that K3O can improve HCD-induced changes in zebrafish body shape.

### 3.2. Lipid Metabolism Treatment Effect of K3O on HCD-induced Larval Zebrafish

In this research, the lipid accumulation was reduced by K3O in the HCD-induced larval zebrafish model. In Figure 1A, Nile red was used to indicate the distribution of the zebrafishes’ lipids in the K3O model and control groups. As shown in the results, the fluorescence intensity and area significantly increased in the model group compared with the control group. However, the intensity and area of the drug-administered groups were weaker than the model group. The larval zebrafishes in the K3O groups exhibited a dose-dependent decrease in neutral lipids, which was reflected by the stained level of Nile red. Moreover, liver lipid accumulation (Figure 2C) was presented by Oil Red stain and HE stain in this study. In the Oil Red stain experiments, K3O supplementation reduced the area of red color in the liver compared with the model group. Moreover, The HE results showed that the lipid reduced in the K3O group when compared with the significant macrovesicular steatosis in the HCD group. We also detected levels of TG and TC in all the larval zebrafish. Figure 2D suggests that both the TG and TC levels were significantly increased by the HCD diet compared with the control group. However, the reduction was more significant in the K3O group, and was also dose-dependent.

In FFA-induced HepG2 cell models, lipid accumulation was remarkable in the model group compared with the control group in the Oil Red stain results. However, the lipid accumulation reduction was more evident in the K3O group than the model group (Figure 3A). These results indicate that K3O has a regulatory effect on lipid metabolism in the larval zebrafish.

### 3.3. Oxidation Stress Treatment Effect of K3O on HCD-induced Larval Zebrafish

ROS plays an important role in oxidation stress [18,19], and is related to lipid overoxidation, oxidant lipid toxins, and inflammation in vivo [20,21]. This is also the case for NAFLD. To investigate the effect that K3O has on oxidation, DCFH-DA was used to stain the ROS as an indicator of oxidant stress in larval zebrafish. As shown in Figure 3A, the fluorescence intensity significantly increased in the fishes’ abdomens in the model group compared with the control group. However, the fluorescence intensity decreased in the BZT and K3O groups. A similar quantified result of ROS was shown in Figure 3B. The MDA level, which illustrates the level of lipid peroxidation, was significantly higher in the HCD group than in the control group (Figure 3C). Additionally, as shown in Figure 3D, the intensity of the GSH-px content was negatively correlated with the MDA. Notably, the K3O group showed a dose-dependent relationship with ROS, MDA, and GSH-px.

The antioxidant effect of K3O was also tested in H_2_O_2_-induced HepG2 cells. DHE staining results showed that the fluorescence of the cells significantly increased in the H_2_O_2_ induced groups compared with the control group. However, in the K3O group, the fluorescence strength significantly decreased (Figure 4B). Moreover, K3O reversed the cells’ reduced viability induced by H_2_O_2_ in the 3-(4,5-Dimethylthiazol-2-yl)-2,5-diphenyltetrazolium bromide (MTT) test (Figure 4C). Meanwhile, the intensity of MDA and GSH-px in the cells was improved in the H_2_O_2_-induced cell injuries in the K3O group, despite the change in level of those indicators being insignificant in some K3O groups (Figure 4D). Still, the improvements were dose-dependent.

In summary, K3O presented a significant antioxidant effect in both the HCD-induced larval zebrafish model and FFA-induced cell model.

### 3.4. Inflammation Treatment Effect of K3O on HCD-induced Larval Zebrafish

Tg (*mpx*: EGFP) zebrafish induced by HCD were used to test the K3O effect on inflammation. As shown in Figure 3D, the abdomen of the zebrafishes underwent a remarkable increase in fluorescence in the HCD-induced group compared with the control group. However, in the K3O and BZT groups, the fluorescence significantly decreased. The quantification of the fluorescence intensity of Tg (*mpx*: EGFP) zebrafish showed a similar effect to the imageology conclusion (Figure 3D). Similarly, the results were dose-dependent in the K3O group.

### 3.5. Mechanism of K3O in Multiregulations

To further reveal the underlying mechanism of K3O on the multiple pathogenesis aspects of NAFLD in zebrafish, we performed an RT-PCR experiment to detect the changes in mRNA expression in lipid generation, lipid oxidation, oxidation stress, fibrosis, and inflammation. As shown in Figure 5A, the expression of the kelch-like ECH-associated protein 1 (*keap1*) gene increased in the model group, and decreased in the K3O group when compared with the control group. In addition, the antioxidant related nuclear fac-tor-like 2 (*nrf2*) gene’s expression increased in the model group but decreased in the K3O group (Figure 5A). Moreover, the expression of the tumor necrosis factor alpha (*tnfa*) inflammation gene and interleukin 6 (*il-6*) significantly increased in the model group, but decreased by K3O compared with the control group. However, the expression of lipid-related genes and fibrosis genes did not show any notable changes in the K3O group when compared with the model group. During the cell experiments, K3O also significantly improved the expression of the Nrf2 and Keap1 gene mRNA compared with the model group (Figure 5B).

In conclusion, K3O exhibited an antioxidant effect by reversing the signal pathway of Nrf2 and Keap1. In addition, K3O improved the expression of inflammation-related genes in the NAFLD model.

## 4. Discussion

The present study suggests that K3O can relieve non-alcoholic fatty hepatitis, and its protective effect may be related to the regulation of Nrf2/Keap1 signals. K3O is found in Ilicis Purpureae Folium and other traditional Chinese medicines, such as Iljinskaja leaves, herba sarcandrae, and artemisia selengensis, etc. [21,22,23,24]. In the HCD-induced larval zebrafish model, K3O can reduce hepatic lipid accumulation. We also used BZT (10 μM), a pan-peroxisome proliferator-activated receptor agonist [22], as a positive drug to compare its lipid regulating effect with K3O. This showed that K3O can reverse the accumulation of abdominal lipids induced by HCD in the liver in the larval zebrafish model. K3O also reduced the increase of TG and TC levels in the larval zebrafish model. However, there were no significant results in the mRNA expression of lipid metabolism (Figure 5) which showed an improvement in lipid generation or lipid elimination related to gene expressions being changed by K3O. This indicates that K3O can regulate other signaling pathways moreover the lipid metabolic pathways. To investigate the direct lipid regulation of K3O, an FFA-induced HepG2 cell model was conducted. Lipids reducing the efficiency of K3O was further observed in the HepG2 cells model. This suggests that K3O can regulate lipid dis-orders in HCD models in vivo and in vitro.

Cholesterol is a major lipid agent associated with oxidation and inflammation. Cholesterol molecules affect the permeability and fluidity of the mitochondria membrane bilayer [23], which is crucial for the transportation of mitochondrial GSH. Meanwhile, intact mitochondrial GSH is essential for maintaining the balance of ROS in mitochondria [24]. Moreover, cholesterol can activate hepatic Kupffer cells, which are the primary source of hepatic proinflammatory cytokines [25], leading to further inflammatory injury to the liv-er.

To further investigate the lipotoxicity caused by cholesterol in oxidative damage, this study investigated the ROS level in an HCD-induced larval zebrafish model. The ROS level in HCD zebrafish was significantly reduced by K3O, which could be seen by a reduction in the fluorescence intensity of DCFH-DA (Figure 3). Interestingly, K3O in the 40 μM group showed a better reduction than BZT in terms of the ROS level. In addition, a significant intensity change of GSH-px was observed in the K3O group. These results indicate that K3O affected antioxidant levels and improved lipid pre-oxidation in vivo. They also suggest that K3O can protect HepG2 from oxidative injury induced by H_2_O_2_ through reducing superoxide levels in vitro. The results accorded with the previous report which stated that K3O could reduce the level of superoxide in vitro [26]. We demonstrated that K3O could regulate ROS disorder and protect liver cells, but the mechanism of the antioxidant effect should be explored. The Nrf2-Keap1 signaling pathway is a key signaling pathway re-sponsible for the antioxidant defense against oxidation [27]. It has been widely reported as a potential treatment target of NAFLD in the liver [28]. Nrf2, as a dominant regulator of the antioxidant response, exists in multiple tissues and protects against oxidation and inflammation [29]. Keap1, acting as a major inhibitory regulator of Nrf2, can bind to Nrf2 and cause it to carry out proteasomal degradation [30]. Therefore, to further reveal whether the antioxidant effect of K3O was mediated by the Nrf2-Keap1 signaling pathway, the Nrf2 and Keap1 mRNA expression levels were investigated. Our results suggest that K3O significantly regulates both Nrf2 and Keap1 expression in vivo and in vitro. This may be the explanation for K3O not presenting directly regulated lipid-related gene expression, and down-regulating lipid accumulation in larval zebrafish.

To further investigate the inflammation protection efficiency of K3O, we conducted a Tg (*mpx*: EGFP) zebrafish model. In Tg (*mpx*: EGFP) zebrafish, the neutrophil-specific protein myeloperoxidase (*mpx*) promoter drives EGFP protein expression in neutrophils [31]. Tg (*mpx*: EGFP) zebrafish can directly reflect the level of neutrophils in vivo experiments, and were used to investigate anti-inflammatory compounds in previous studies [25]. Recently, anti-inflammatory components of PSORI-CMo2 were selected by Tg (*mpx*: EGFP) in Yuan’s research [32]. In the present study, a significant anti-inflammatory effect of K3O (40 μM) was observed in HCD-induced zebrafish. The following test revealed that it was associated with decreased mRNA expression on the proinflammatory tnfα and il6 (Figure 5), which further indicates the anti-inflammatory effects of K3O.

Due to it being a complex disease, recent studies believed that the mechanism of NAFLD is related to multiple aspects, including endoplasmic reticulum stress [33], insulin resistance [34], autophagy [35], fibrosis [36], and gut-liver axis [37]. Even though this paper investigates the K3O effect on lipid disorder, oxidation, and inflammation, studies investigating the effect of K3O on other mechanisms of NAFLD are still lacking. It was found that K3O compounds exist in many kinds of Chinese herbs, offering different levels of efficacy, and in the pharmacokinetic study of related Chinese herbs, K3O had a higher blood concentration and longer half-life. This means that K3O may be more likely to enter the body through the intestinal barrier to exert its therapeutic effect [38,39]. Therefore, pharmacokinetic studies of K3O will be a potential direction in further research. Since K3O is a glycosidic structure, whether there is a biological transformation of K3O in the parent nucleus intestine and in vivo should be more focused on. These questions will thus be addressed in subsequent experimental studies.

So far, the significant antioxidant effect of K3O has been revealed, it also suggests to us that more studies involving K3O’s effects on the treatment and mechanism in other aspects of NAFLD are worth investigating. Based on all the above results, this study revealed the lipid regulation, antioxidant and anti-inflammatory effects of K3O in the HCD-induced NAFLD larval zebrafish model, and provided an understanding of K3O’s lipid regulation, antioxidant, and anti-inflammatory effects for the advancement of therapeutics to treat NAFLD.

## 5. Conclusions

In summary, K3O demonstrated an ability to regulate lipid oxidation and antioxidant effects in both the larval zebrafish model and HepG2 cell model. This study showed that K3O can simultaneously protect cells from liver damage caused by H_2_O_2_, reduced MDA, and enhanced GSH-px levels in the NAFLD zebrafish model. The RT-PCR results indicate that the expression of Nrf2/Keap1 was changed by K3O in larval zebrafish. Changes in the levels of mRNA in the same genes were also observed in cell experiments after K3O treatment. All results lead to the conclusion that K3O fulfilled its function in the NAFLD zebrafish model. Although the exact mechanism of K3O remains to be discovered, the present study supports the potential of K3O to be developed into an anti-NAFLD drug in further studies.

## Figures and Tables

**Figure 1 life-11-00445-f001:**
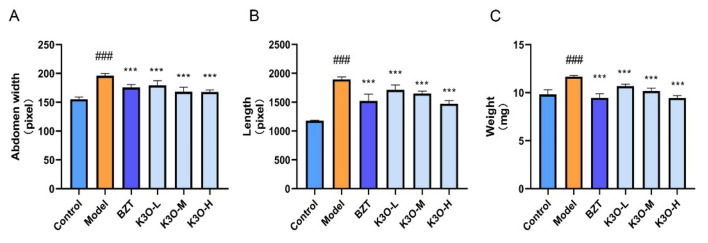
Effect of K3O on the body size of HCD-induced larval zebrafish. (**A**) Abdomen width of larval zebrafish. (**B**) Length of larval zebrafish. (**C**) Weight of larval zebrafish livers. The bars indicate mean ± SD. n.s. *p* > 0.05; # *p* < 0.05, ## *p* < 0.01, ### *p* < 0.001 represent the difference in significance compared with control; *p* > 0.05, * *p* < 0.05, ** *p* < 0.01, and *** *p* < 0.001 represent the difference in significance compared with model, *p* < 0.05 was considered to be statistically significant. Significance was calculated by ANOVA followed by Turkey’s test (*n* = 10 for **A**–**C**).

**Figure 2 life-11-00445-f002:**
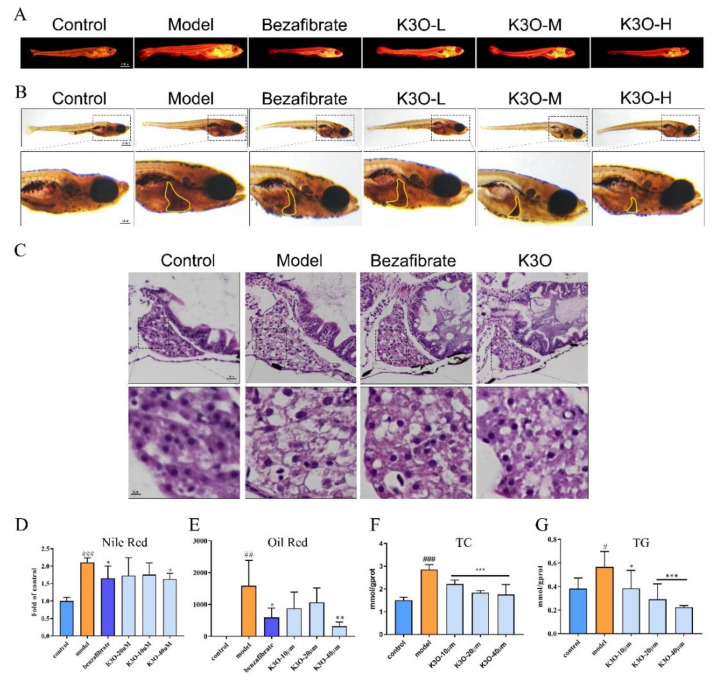
Effect of K3O on the lipid accumulation of HCD-induced larval zebrafish. (**A**) Nile red stain of larval zebrafish. (**B**) Oil Red stain of larval zebrafish; the hepatic steatosis is indicated by the yellow circle. (**C**) HE stains of larval zebrafish livers. (**D**) Quantitation of Nile red stain. (**E**) Quantitation of Oil Red stain. (**F**) TC level of larval zebrafish. (**G**) TG level of larval zebrafish. The bars indicate mean ± SD. n.s. *p* > 0.05; # *p* < 0.05, ## *p* < 0.01, ### *p* < 0.001 represent the difference of significance compared with control; * *p* < 0.05, ** *p* < 0.01, and *** *p* < 0.001 represent the difference of significance compared with model, *p* < 0.05 was considered to be statistically significant. Significance was calculated by ANOVA followed by a Turkey’s test (*n* = 10 for **D** and **E**; *n* = 18 in three separate runs for **F** and **G**).

**Figure 3 life-11-00445-f003:**
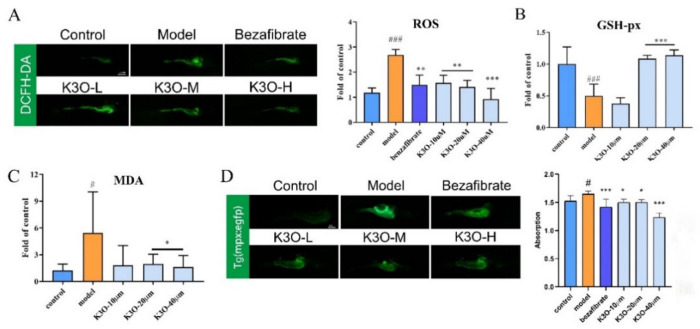
Effect of K3O on the oxidation and inflammation of HCD-induced larval zebrafish. (**A**) ROS of larval zebrafish stained by DCFH-DA and quantification. (**B**) GSH-px level of larval zebrafish. (**C**) MDA level of larval zebrafish. (**D**) Tg (*mpx*: EGFP) zebrafish captured by a fluorescence stereoscope and fluorescence intensity quantification. The bars indicate mean ± SD. n.s. *p* > 0.05; # *p* < 0.05, ## *p* < 0.01, ### *p* < 0.001 represent the difference of significance compared with control; * *p* < 0.05, ** *p* < 0.01, and *** *p* < 0.001 represent the difference of significance compared with model, *p* < 0.05 was considered to be statistically significant. Significance was calculated by ANOVA followed by a Turkey’s test (*n* = 10 for **A** and **D**; *n* = 18 in three separate runs for **B** and **C**).

**Figure 4 life-11-00445-f004:**
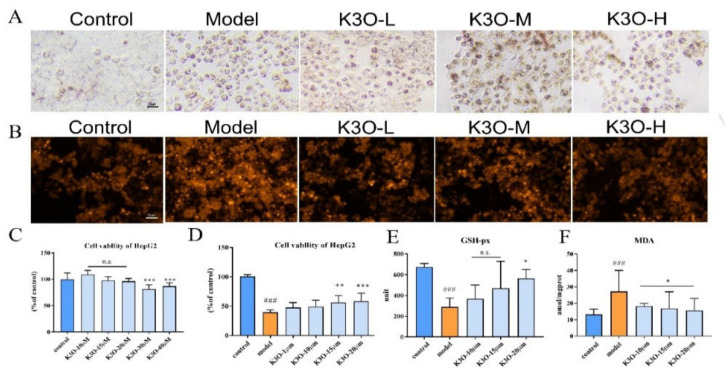
Effect of K3O on FFA-induced HepG2 cell line. (**A**) Oil Red stain of HepG2. (**B**) DHE stain of HepG2. (**C**) HepG2 cell viability of K3O detected by MTT. (**D**) H_2_O_2_-induced HepG2 cell viability of K3O detected by MTT. (**E**) GSH-px level of HepG2. (**F**) MDA level of HepG2. The bars indicate mean ± SD. n.s. *p* > 0.05; # *p* < 0.05, ## *p* < 0.01, ### *p* < 0.001 represent the difference of significance compared with control; * *p* < 0.05, ** *p* < 0.01, and *** *p* < 0.001 represent the difference of significance compared with model, *p* < 0.05 was considered to be statistically significant. Significance was calculated by ANOVA followed by a Turkey’s test (*n* = 18 in three separate runs for **C** and **D**; *n* = 3 in three separate runs for **E** and **F**).

**Figure 5 life-11-00445-f005:**
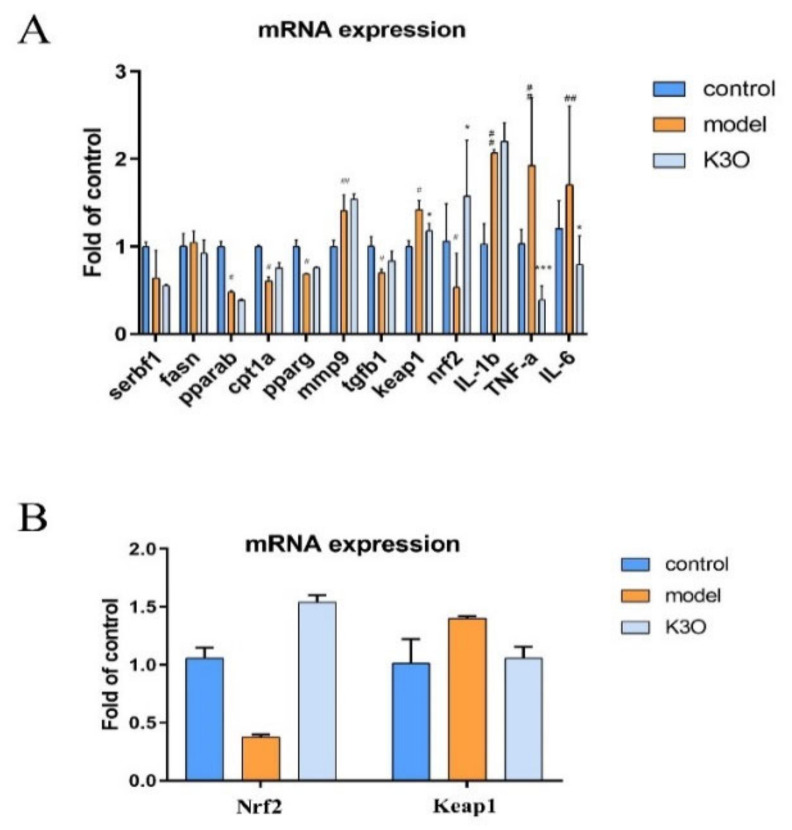
mRNA expression effect of K3O. (**A**) mRNA expression of larval zebrafish. (**B**) Nrf2 and Keap1 mRNA expression in HepG2. The bars indicate mean ± SD. n.s. *p* > 0.05; # *p* < 0.05, ## *p* < 0.01, ### *p* < 0.001 represent the difference of significance compared with control; * *p* < 0.05, ** *p* < 0.01, and *** *p* < 0.001 represent the difference of significance compared with model, *p* < 0.05 was considered to be statistically significant. Significance was calculated by ANOVA followed by a Turkey’s test. (*n* = 60 in three separate runs for **C** and **D**; *n* = 3 in three separate runs for **E** and **F**).

**Table 1 life-11-00445-t001:** Specific sequences of primers used in RT-qPCR. The specific sequences of the primers used in this study are shown in this table. The Danio rerio primers were used on larval zebrafish. Equivalently, Homo sapiens primers were listed for reference.

Gene Name	Species	Forward Primer (5′ -> 3′)	Reverse Primer (5′ -> 3′)
serbf1	Danio rerio	CATCCACATGGCTCTGAGTG	CTCATCCACAAAGAAGCGGT
fasn pparb cpt1a pparg mmp9 tgfb1 keap1 nrf2 il1b tnfa	Danio rerio Danio rerio Danio rerio Danio rerio Danio rerio Danio rerio Danio rerio Danio rerio Danio rerio Danio rerio	ATCTGTTCCTGTTCGATGGC CGTCGTCAGGTGTTTACGGT ACTCTCGATGGACCCTGTGA CTGCCGCATACACAAGAAGA GAAGCGTTACGGCTACGT CATAAGAGCCACAGACAGAAG CCAACGGCATAGAGGTAGTTAT TTGTCTTTGGTGAACGGAGGT TGGCGAACGTCATCCAAG GCTTATGAGCCATGCAGTGA	AGCATATCTCGGCTGACGTT AGGCACTTCTGGAATCGACA CTGGATGAAGGCATCTGGAC TCACGTCACTGGAGAACTCG TTCCATGTCTGGCGAATAG GTAGAGCGAGCGTAAACAG CCTGTATGTGGTAGGAGGGTT CTCGGAGGAGATGGAAGGAAG GGAGCACTGGGCGACGCATA TGCCCAGTCTGTCTCCTTCT
il6	Danio rerio	AGACCGCTGCCTGTCTAAAA	TTTGATGTCGTTCACCAGGA
Nrf2	Homo sapiens	ACCTCCCTGTTGTTGACTT	CACTTTATTCTTACCCCTCCT
Keap1	Homo sapiens	TTACGACCCAGATACAGACA	TGCCCAAGAAACAAAAGT

## Data Availability

Not applicable.

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
