# Peer review of "Kaempferol-3-O-Glucuronide Ameliorates Non-Alcoholic Steatohepatitis in High-Cholesterol-Diet-Induced Larval Zebrafish and HepG2 Cell Models via Regulating Oxidation Stress"

_life, 2021, doi:10.3390/life11050445_

Round 1
Reviewer 1 Report
As one of the most common and rapidly-growing diseases in the US and worldwide, non-alcoholic fatty liver disease (NAFLD) currently presents an immense public health burden. Despite this, our understanding of NAFLD pathophysiology is very limited, therefore, limiting our treatment options for this disease.
In this study, Deng et al. investigate the effects of Kaempferol-3-O-glucuronide (K3OG) on oxidative stress, lipid metabolism, and inflammation in a zebrafish model organism fed high cholesterol diet or HepG2 cell model treated with free fatty acid or H2O2. These findings provide insights into the anti-oxidative and anti-inflammatory properties of K3OG that could be utilized for developing an improved NAFLD treatment.
Overall, the studies in this manuscript are well-designed and are described in sufficient detail. The results are clearly described, although the data presentation can be improved (see my comments below). The discussion and conclusions need to be modified to support the findings presented in the manuscript.
Specifically, I would like the authors to address the following comments/concerns:
- Figure 1 and Table 1 should be moved to Supplementary material. Also, the formatting of Table 1 should change since it is currently unclear what the multiple columns show.
- In Figure 2A, are all these fish the same age or imaged at the same magnification? There seems to be a significant difference in size between the model and all the treated conditions. Similarly, in 2B, the liver size looks different between the model and the treated conditions. Please comment on this.
- Rather than the term “model” please use the exact treatment in both zebrafish and HepG2 cells. It will help clarify more quickly what the model is.
- In Figure 5A, the gene name needs to be changed from serbf1 to srebf1. Figure 5B is missing the significance marks.
- The final statement in Results needs to be changed: “In conclusion, K3GO performed the antioxidant effect by reversing the signal pathway of Nrf2 and Keap1.” This statement cannot be substantiated by the data included in this manuscript, which is correlative at the moment.
I would be happy to evaluate this manuscript again and accept it for publication, given the appropriate response to my concerns listed above.
Author Response
Reviewer #1:
- Response to comment: Figure 1 and Table 1 should be moved to Supplementary material.
Response: Figure 1 and Table 1 are moved to supplementary materia and Table 1 is changed to a clearer version of the Excel version for ease of reference.
- Response to comment: In Figure 2A, are all these fish the same age or imaged at the same magnification?
Response: All zebrafish used in the experiment were photographed at the same age and at the same magnification. We check the data of zebrafish body size and found that HCD induced larval zebrafish body length and abdominal width. For this change, we compiled the data of the length, abdominal width, and weight of the zebrafish in the experiment, and added this data to the manuscript, which shows the results in Figure1.
- Response to comment: Rather than the term “model” please use the exact treatment in both zebrafish and HepG2 cells. It will help clarify more quickly what the model is.
Response: Thank you very much for your suggestion. We have revised and corrected the sentences in the article according to your suggestion. The corrected ones are shown in blue in the manuscript.
- Response to comment 4 and comment 5: Figure 5A, 5B error and the final statement in results needs to be changed.
Response: Thank you so much for your suggestion, and we are check the srebf gene name and rewrite the wrong writing. Figure 5B significant markers have been added. Based on suggestion, we have revised the conclusion and sign the sentence in blue color.
Reviewer 2 Report
In this manuscript, Deng et al. present results showing the ameliorating effect of Kaempferol-3-O-glucuronide treatment in an in vitro (human hepatic cell culture) and in vivo ( zebrafish) model of NASH. The manuscript is interesting and can be accepted for publication after a major revision.
1: The main problem this reviewer found in following the manuscript was the connection between beneficial effect of pure K3GO on health and Illicis Purpureaea Folium traditional use. What is the amount of K3GO present in the plant? Does its amount justify the beneficial effect of the plant? What is the pharmacokinetics of K3GO? Is it metabolized by the microbiota? Is it transformed into other active metabolites? Considering that the active concentration registered by the authors is in the micromolar range, how much Ilicis one should eat to reach the active concentration of K3GO in human serum?
2: the second problem is the network pharmacology part, which, in my opinion, is the weakest part of the manuscript. Indeed, the same target disease mechanism suggestion would have resulted in interrogating the software against many other natural compounds. I think this part could be removed or downscaled in terms of importance.
Author Response
Reviewer #2:
- Response to comment: The main problem this reviewer found in following the manuscript was the connection between beneficial effect of pure K3GO on health and Illicis Purpureaea Folium traditional use. What is the amount of K3GO present in the plant? Does its amount justify the beneficial effect of the plant? What is the pharmacokinetics of K3GO? Is it metabolized by the microbiota? Is it transformed into other active metabolites? Considering that the active concentration registered by the authors is in the micromolar range, how much Ilicis one should eat to reach the active concentration of K3GO in human serum?
Response: Thank you very much for your questions. They are very valuable and should be taken seriously in traditional herbal medicine. To answer your questions, I have consulted the relevant English and Chinese literature. K3O is found in a variety of natural medicines, such as holly leaves, Cyclocarya paliurus [1], snow lotus [2], medlar leaves [3], Herba Sarcandrae [4] and so on. Unfortunately, only in the literature on holly leaves were we able to find K3O concentrations of 1.33-0.51mg/g relative to the plant [5]. The content data of K3O in other plants was not found. Herba Sarcandrae's literature indicates that after Herba Sarcandrae was given, K3O levels were detected in rats with AUC0-t of 1643.5 ng h, ml and Cmax of 194.6 and 179.5, respectively, with a half-life of 3.89 h [6]. Although K3O may be transfected into Herba Sarcandrae, it remains to be seen whether the amount of the transfected drug would have a beneficial effect. The pharmacokinetics of K3O alone, its effect on intestinal flora and its transformation in the intestine and in vivo have not been reported in detail, which will be the focus of our future work. And we already dicussion the related question in discussion, the re-writing part were present in blue in the manuscript.
- Response to comment: The second problem is the network pharmacology part, which, in my opinion, is the weakest part of the manuscript.
Response: Based on your suggestion, we have revised the network pharmacology section. The web pharmacology section has been narrowed down in the article and is only mentioned in methodologies and extraneous materials.
Reviewer 3 Report
The manuscript is well structured and clear. The study turns out to be very interesting, the conclusions are supported by the results and with important clinical repercussions. However, proofreading in English is recommended.
The aim has been partially achieved as the explanation of the molecular mechanism of action of the K3OG compound is lacking. Furthermore, the NAFLD model used (high cholesterol diet) is not the only one validated for zebrafish. Why was the HCD diet used and not the High Fructose diet, which seems to cause more steatosis and more oxidative damage?
Author Response
Reviewer #3:
- Response to comment: proofreading in English is recommended.
Response: Thank you very much for your questions. We reworked the full language and found a local language scholar to modify the article.
- Response to comment: Why was the HCD diet used and not the High Fructose diet, which seems to cause more steatosis and more oxidative damage?
Response: Thank you very much for your advice. Before starting the experiment, we compared the effects of several models on zebrafish, including a diet high in sugar and cholesterol. We also carried out experiments on high sugar molding, and found that high sugar content changed the viscosity and quality of water. The water viscosity and the high sugar model characteristic, the water quality easy to breed the bacterium, has caused the bigger influence to zebrafish survival environment. But HCD feed is a solid feed, which has little effect on the viscosity of water. So we choose HCD model to induce NAFLD from the point of view of experimental technology.
Round 2
Reviewer 2 Report
The manuscript can be accepted for pubblication
This manuscript is a resubmission of an earlier submission. The following is a list of the peer review reports and author responses from that submission.